# Genotypic and Phenotypic Characterization of Pathogenic *Escherichia coli*, *Salmonella* spp., and *Campylobacter* spp., in Free-Living Birds in Mainland Portugal

**DOI:** 10.3390/ijerph20010223

**Published:** 2022-12-23

**Authors:** Rita Batista, Margarida Saraiva, Teresa Lopes, Leonor Silveira, Anabela Coelho, Rosália Furtado, Rita Castro, Cristina Belo Correia, David Rodrigues, Pedro Henriques, Sara Lóio, Vanessa Soeiro, Paulo Martins da Costa, Mónica Oleastro, Angela Pista

**Affiliations:** 1Food Microbiology Laboratory, Food and Nutrition Department, National Institute of Health Doutor Ricardo Jorge, Avenida Padre Cruz, 1649-016 Lisbon, Portugal; 2Food Microbiology Laboratory, Food and Nutrition Department, Rua Alexandre Herculano 321, 4000-055 Oporto, Portugal; 3National Reference Laboratory for Gastrointestinal Infections, Department of Infectious Diseases, National Institute of Health Doutor Ricardo Jorge, Avenida Padre Cruz, 1649-016 Lisbon, Portugal; 4ESAC-IPC, Coimbra College of Agriculture, Polytechnic of Coimbra, 3045-601 Coimbra, Portugal; 5CEF, Forest Research Centre, Edifício Prof. Azevedo Gomes, ISA, Tapada da Ajuda, 1349-017 Lisboa, Portugal; 6Espaço de Visitação e Observação de Aves, 2600 Vila Franca de Xira, Portugal; 7Centro de Recuperação de Fauna do Parque Biológico de Gaia, Rua da Cunha, Avintes, 4430-812 Vila Nova de Gaia, Portugal; 8ICBAS—Institute of Biomedical Sciences Abel Salazar, Rua de Jorge Viterbo Ferreira, 228, 4050-313 Oporto, Portugal

**Keywords:** *Campylobacter* spp., *Salmonella* spp., pathogenic *Escherichia coli*, whole-genome sequencing, free-living birds, Portugal

## Abstract

Birds are potential carriers of pathogens affecting humans and agriculture. Aiming to evaluate the occurrence of the top three most important foodborne pathogens in free-living birds in Portugal, we investigated 108 individual fecal samples from free-living birds and one pooled sample of gull feces (*n* = 50) for the presence of *Escherichia coli* (pathogenic and non-pathogenic), *Salmonella* spp. and *Campylobacter* spp. Virulence- and antimicrobial resistance- (AMR) associated genes were detected by PCR and Whole-Genome Sequencing (WGS), and phenotypic (serotyping and AMR profiles) characterization was performed. Overall, 8.9% of samples tested positive for pathogenic *E. coli*, 2.8% for *Salmonella* spp., and 9.9% for *Campylobacter* spp. AMR was performed on all pathogenic isolates and in a fraction of non-pathogenic *E. coli*, being detected in 25.9% of them. Ten of the tested *E. coli* isolates were multidrug-resistant (MDR), and seven of them were Extended-spectrum β-lactamase (ESBL) producers. Among *Salmonella* (*n* = 3) and *Campylobacter* (*n* = 9), only one strain of *C. jejuni* was identified as MDR. Most of the identified serotypes/sequence types had already been found to be associated with human disease. These results show that free-living birds in Portugal may act as carriers of foodborne pathogens linked to human disease, some of them resistant to critically important antimicrobials.

## 1. Introduction

Many emerging and reemerging diseases that affect human health are caused by pathogens that have once emerged from an animal and later have crossed the species barrier to infect humans. Approximately 60% of all human infectious diseases recognized so far, and about 75% of emerging infectious diseases, are of zoonotic origin [1].

The dynamic interaction between humans, animals, and pathogens sharing the same environment makes the One Health approach essential for the identification and control of emergent zoonotic bacterial diseases. Collaborative international surveillance strategies, quick and trustworthy agent identification techniques, and optimization of the responses to zoonotic events will ensure the prevention and management of such infections [2].

According to the “The European Union One Health 2020 Zoonoses Report” of the European Food Safety Authority (EFSA) and the European Centre for Disease Prevention and Control (ECDC), campylobacteriosis, salmonellosis, and Shiga toxin-producing *E. coli* (STEC) infections were on the top four of the most reported zoonoses in humans in 2020, representing about 95% of all the confirmed human cases and involving over 15,000 hospitalizations [3].

Wild birds can become reservoirs and/or mechanical vectors for these pathogenic bacteria and for the antimicrobial-resistant genes they carry [4,5,6,7,8,9]. Throughout the time, many of these species have successfully adapted to human-dominated environments and regularly come into close contact with livestock, domestic animals, and people [10]. Furthermore, many of these species are migratory, flying thousands of kilometers to find the best ecological conditions and habitats for feeding, breeding, and raising their young, which can also contribute to widespread dissemination [11,12].

The transmission of these pathogens to humans can occur by different routes, either by direct or indirect contact, for example, through the contamination of soil, water, or agricultural food supplies [13].

A better comprehension of wild birds’ migration routes, as well as further research on the prevalence, pathogenesis, and clinical associations of the different migratory birds’ pathogens, will be crucial to a better understanding of the disease transmission dynamics, helping to prevent future outbreaks of relevant human infections. 

Limited information exists about the occurrence and characterization of zoonotic bacteria isolated from wild birds throughout Europe [14,15,16,17,18,19,20,21,22]. This information is even scarcer if focusing on Portugal [23,24,25,26,27,28,29,30]. Therefore, our aim was to examine the occurrence of pathogenic *E. coli*, *Salmonella* spp., and *Campylobacter* spp. in fecal samples collected from free-living birds from different regions of Portugal. Furthermore, the serotypes/sequence types, associated virulence markers, and antimicrobial resistance (AMR) profiles of these major foodborne bacteria reported in Europe in 2020 were also characterized. Considering that the microbiomes of wild birds could act as efficient AMR reservoirs and epidemiological links among humans, livestock, and natural environments [31], the AMR profile of the majority of the non-pathogenic *E. coli* was also attained.

This is the first study in Portugal, as far as we know, and one of the few in Europe, that concurrently evaluates the presence and performs the characterization of *Campylobacter* spp., *Salmonella* spp., and *E. coli* (STEC and non-STEC) in wild birds by Whole Genome Sequencing in parallel with phenotypic methods.

## 2. Materials and Methods

### 2.1. Studied Areas

This work covered three different geographical areas, one in the Lisbon district and two in the north of Portugal. 

Located in the Lisbon district, in the heart of the most important wetland in Portugal, the Tagus Estuary Natural Reserve, EVOA (Espaço de Visitação e Observação de Aves) integrates three freshwater wetlands, summing up to a total of 70 ha. These lagoons play an important role in birdlife, being used as both a refuge and nesting areas. EVOA’s main motivation is the conservation of birdlife and its scientific study.

The Parque Biológico de Gaia (PBG) is a 35 ha park situated in an agro-forestry area in the Vila Nova de Gaia municipality in the north of Portugal, and it shelters hundreds of species in their habitats. It holds a Wildlife Rehabilitation Center (Centro de Recuperação de Fauna do Parque Biológico de Gaia- CRFPBG) were injured, debilitated, or illegally kept animals are treated and, whenever possible, returned into the wild.

The Oporto coastline has a large gull population composed of residents (*Larus michahellis*) and migratory species (*L. fuscus* and *L ridibundus*), which can range the UK, Germany, the Netherlands, and Scandinavia [29,32,33]. 

### 2.2. Studied Population and Sample Collection

Fecal samples from free-living birds (*n* = 108) from at least 30 species from 10 different orders were collected by swirling cotton swabs in birds’ fresh feces, right after defecation, between September 2020 and June 2021. At EVOA, fecal samples were gathered between September to November 2020 during ringing sessions, while the CRFPBG samples were collected from hosted animals between November 2020 and June 2021. All samples were refrigerated in a cool bag and processed immediately after arriving at the laboratory.

In addition, seven *E. coli* isolates recovered from wild gulls at Instituto de Ciências Biomédicas Abel Salazar (ICBAS) were included in this study. One pooled sample made up of 50 individual fecal droppings was collected in July 2021 from Matosinhos beach (Oporto coastline) in the Portuguese northwest region, using a sterile glove. The samples were taken at dawn when gulls are still on the beach. Care was taken during sampling to avoid the collection of beach sediment. All fecal samples were processed in the laboratory within 2 h.

### 2.3. Isolation Methodology

Isolation of *E. coli* was performed as previously described [34]. Briefly, a pre-enrichment step was followed by plating-out on selective and non-selective media. Confirmation of suspicious colonies was achieved on VITEK 2 compact system (bioMérieux, Marcy L’Etoile, France) or by *E. coli* 16S rRNA amplification (Sabat et al., 2000) [35]. 

*Salmonella* spp. isolation was based on ISO 6579-1:2017 [36], as previously described [34]. Briefly, after a non-selective and selective enrichment and plating-out, colonies of presumptive *Salmonella* spp. were isolated on Tryptone Soy Agar (TSA. Biokar Diagnostics). VITEK 2 compact system was used for the identity confirmation of the suspicious isolates. 

*Campylobacter* spp. was isolated according to ISO 10272-1:2006-1 [37], as previously described [34]. After several enrichment and incubation steps in a microaerobic atmosphere, presumed *Campylobacter* colonies were sub-cultured on Columbia Agar + 5% Sheep blood (COS, bioMérieux) and confirmed at the genus and species level by the oxidase activity test and MALDI-TOF mass spectrometry, respectively (VITEK^®^ MS, bioMérieux). 

### 2.4. Typing and Antimicrobial Susceptibility Testing (AST)

Pathogenic *E. coli* identification was based on the presence of the most common virulence genes (VG), *eae*, *aggR*, *elt*, *estp*, and *ipaH*, performed by multiplex PCR (modified from [38,39,40]) and on the presence of Shiga toxins genes *stx1* and *stx2* [41]. Pools with up to ten colonies from each sample were initially considered, each isolate was analyzed individually whenever a PCR-positive result was detected, as previously described [34]. Appendix A presents primer sequences and PCR profiles. Potentially pathogenic isolates (STEC; EPEC, Enteropathogenic *E. coli*; EAEC, Enteroaggregative *E. coli*; ETEC, Enterotoxigenic *E. coli*; EIEC, Enteroinvasive *E. coli*) were characterized by the presence of at least one of the tested genes. 

AST was carried out by disc diffusion in a total of 54 isolates, 5 pathogenic *E. coli* (PCR-based) and 49 presumptive non-pathogenic *E. coli*, for 18 antimicrobials (Ampicillin, Amoxicillin-Clavulanic Acid, Azithromycin, Cefepime, Cefotaxime, Cefoxitin, Ceftazidime, Ceftriaxone, Chloramphenicol, Erythromycin, Gentamicin, Meropenem, Nalidixic Acid, Ciprofloxacin, Sulfamethoxazole, Tetracycline, Tigecycline, and Trimethoprim). This assay was performed following the European Committee on Antimicrobial Susceptibility Testing (EUCAST) recommendations [42], and EUCAST epidemiological cut-off values (ECOFFs) were used for the results interpretation. Multidrug resistance (MDR) was defined as resistance to three or more antimicrobial classes. 

Serotyping of *Salmonella* was performed in accordance with the Kauffmann-White-Le Minor scheme [43] by slide agglutination for O and H antigens (SSI Diagnostica, Hillerod, Denmark; Sifin diagnostics, Berlin, Germany). AST followed the same panel of antimicrobials for *E. coli*, excluding Ciprofloxacin and Erythromycin and including Pefloxacin.

Regarding *Campylobacter* spp., AST was accomplished for Ciprofloxacin, Erythromycin, Tetracycline, and Gentamycin according to EUCAST 2021 [42] and for Ampicillin and Amoxicillin-Clavulanic Acid in accordance with the Comité de l’antibiogramme de la Société Française de Microbiologie [44]. 

### 2.5. Whole Genome Sequencing and Genome Characterization

As previously described [34], genomic DNA was extracted from all pathogenic isolates, as well as from all strains classified as MDR. The NexteraXT library preparation protocol (Illumina, San Diego, CA, USA) was used for DNA preparation prior to cluster generation and sequencing on a MiSeq or a NextSeq 550 instrument (Illumina). FastQC v0.11.5 “http://www.bioinformatics.babraham.ac.uk/projects/fastqc/ (accessed on 10 November 2022)” was used for read quality analysis, and Trimmomatic v0.36 (with automatically determined trimming criteria) [45] for read trimming. Sequence reads of *E. coli* and *Salmonella* spp. were submitted to several services present in the Center for Genomic Epidemiology web server “https://cge.cbs.dtu.dk (accessed on 10 November 2022)”. ResFinder 4.1 was used for the identification of AMR genes, and MLST 2.0 for in silico Sequence Type (ST). Furthermore, VirulenceFinder 2.0 allowed the identification of *E. coli* virulence genes, and SerotypeFinder 2.0 and SeqSero 1.2 the in silico *E. coli* and *Salmonella* serotyping, respectively. 

For *Campylobacter*, in silico Multilocus Sequence Typing was achieved at the PubMLST platform “https://pubmlst.org/ (accessed on 10 November 2022)”, and in silico AMR was determined using AMRFinderPlus “https://www.ncbi.nlm.nih.gov/pathogens/antimicrobial-resistance/AMRFinder/ (accessed on 10 November 2022)”. Genomes were annotated using RAST “https://rast.nmpdr.org/ (accessed on 10 November 2022)”, and together with BLAST analysis “https://blast.ncbi.nlm.nih.gov/Blast.cgi (accessed on 10 November 2022)”, a search for type IV and type VI secretion systems (T4SS and T6SS, respectively) was performed, using complete systems available in NCBI database (accession numbers AF226280.2 and JX436460.1 for T4SS and T6SS, respectively). 

Sequence reads were deposited under bioprojects PRJEB54735 (*E. coli*), PRJEB32515 (*Salmonella*), PRJEB46750, and PRJEB46733 (*C. jejuni* and *C. coli*, respectively) in the European Nucleotide Archive (ENA). Appendix A lists accession numbers for each isolate.

## 3. Results

### 3.1. Detection and Characterization of Isolates

In this work, 108 individual fecal samples collected from free-living birds of species belonging to 10 different orders and one pooled sample of gull feces (*n* = 50) were tested for the presence of *Campylobacter* spp., *Salmonella* spp., and *E. coli*.

In total, pathogenic *E. coli*, *Salmonella* spp., and *Campylobacter* spp. were detected in 8.9% (14/158), 2.8% (3/108), and 9.9% (9/91) of the samples, respectively. In addition, it was possible to isolate 54 non-pathogenic *E. coli* (without the presence of *eae*, *aggR*, *elt*, *estp*, *ipaH*, *stx1*, or *stx2*) (34.1%) (Table 1).

Regarding the pathogenic *E. coli*, the distribution was as follows: ExPEC was the most frequently detected pathotype (64.3%, 9/14), followed by STEC and EAEC (14.3% each, 2/14), and EPEC (7.1%, 1/14). Pathogenic *E. coli* was detected in samples from gulls (*Larus* spp.), reed warblers (*Acrocephalus scirpaceus*), common moorhen (*Gallinula chloropus*), teal (*Anas crecca*), and feral pigeon (*Columba livia*) (Table 1).

Extraintestinal pathogenic *E. coli* (ExPEC) were classified based on WGS data analysis. The presence of two or more ExPEC virulence genes was used for this pathotype classification [46]. These isolates were sequenced due to their multidrug resistance (MDR) phenotype. 

Regarding *Campylobacter* species, *Campylobacter* coli was detected in four samples, two isolates from eurasian coots (*Fulica atra*), one from common moorhens (*Gallinula chloropus*), and one from a reed warbler (*Acrocephalus scirpaceus*); *Campylobacter jejuni* was detected in five samples, four from gulls (*Larus* spp.) and one from common moorhens (*Gallinula chloropus*) (Table 1).

*Salmonella* spp. was less frequent in the studied population (2.8%, 3/108), although serovars diversity was observed: one case of *S.*60:k:e,n,x,z15, in a sample of a short-toed snake eagle (*Circaetus gallicus*), and *S.* Typhimurium and *S.* Litchfield in samples from two feral pigeons- (*Columba livia*) (Table 1)

Only one of the 108 analyzed individual fecal samples contained two different pathogens: a common moorhen (*Gallinula chloropus*) sample containing both EAEC and *C. coli*.

### 3.2. Serotyping, Virulence Genes, and MLST Analysis

Amongst the 14 *E. coli* pathogenic isolates (two EAEC, two STEC, one EPEC, and nine ExPEC), 9 O antigens (O25, O45, O55, O78, O92, O101, O105, O111, O125ac) and 9 H antigens (H2, H4, H6, H7, H9, H10, H16, H21, H33) were identified (Table 2).

In addition to genes defining pathotypes, the WGS data analysis revealed the presence of other virulence-associated genes in the 14 pathogenic *E. coli* isolates, such as glutamate decarboxylase gene (*gad*), essential to bacterial persistence in strongly acidic habitats and *terC*, involved in tellurite resistance. Both these genes were present in 100% of the pathogenic *E. coli* studied strains. Other identified *E. coli* virulence genes included the ones encoding transfer protein *TraT* (71.4%), which reduces the sensitivity of the bacterial cells to phagocytosis by macrophages; outer membrane protein *OmpT* (64.3%) that plays a multifaceted role in *E. coli* pathogenesis, increasing bacterial adhesiveness/invasiveness; increased serum survival-*iss* (78.6%), involved in the survival to the bactericidal action of the serum, and siderophores *fyua*, *irp2*, *iutA* (57.1%), *iucC* (57.1%) and *sit A* (42.9%) involved in the acquisition of iron. Adhesin genes encoding long polar fimbriae-*lpfA* and nonhemagglutinin adhesion- *iha* were also present in 42.9%, and 35.7% of the pathogenic *E. coli* isolates, respectively. Finally, toxin genes encoding heat-stable enterotoxin 1- *ast* A and microcin F- *mchF* were likewise present in 35.7% and 28.6% of the 14 isolates, respectively (Table 3), while the *stx2* gene was only present in the two STEC isolates.

It is worth remarking that the isolate encoding more virulence factors was the Extended-spectrum β-lactamase-producing EAEC O111:H21.

In silico MLST analysis identified 11 STs among the 14 *E. coli* pathogenic isolates (Table 2), including a novel sequence type, submitted to Enterobase and assigned as ST13581. EAEC isolates belong to ST34 and ST40; STEC to ST20 and the new ST13581; EPEC to ST583, and ExPEC to ST10, ST23, ST131, ST162, ST410 and ST453. 

The three *Salmonella* isolated in this study were characterized as belonging to ST3127 (*S*.60:k:e,n,x,z15), ST19 (*S*. Typhimurium), and ST214 (*S*. Litchfield). 

*C. coli* MLST analysis retrieved ST11400 and ST11401 among two isolates. ST was not possible to identify in two of the *C. coli* isolates owing to fragmented DNA. *C. jejuni* isolates were characterized as belonging to ST990, ST1268, and ST8572. Moreover, ST was not possible to identify in two of the five *C. jejuni* isolates due to fragmented DNA. The secretion apparatus T4SS and T6SS play a role in both bacterial competition and virulence, and their presence was assessed in *Campylobacter* isolates. Overall, three among five *Campylobacter* isolates analyzed by WGS harbor at least one of these systems: one *C. coli* isolate from *Acrocephalus scirpaceus* harbors gene clusters encoding both T4SS (located in the pVir plasmid) and T6SS, while another *C. coli*, from a *Fulica atra*, and one *C. jejuni* from a gull (*Larus* spp.) were found to carry a complete T6SS.

### 3.3. Antimicrobial Resistance

Through WGS analysis of 54 *E. coli* isolates, antimicrobial resistance (AMR) associated determinants were detected in 14 (25.9%), 11 of which were pathogenic and three non-pathogenic. Ten of the AMR isolates were classified as MDR (10/54, 18.5%), all of them pathogenic, and seven Extended-spectrum β-lactamase (ESBL) producing *E. coli* (Table 3). The phenotypic resistance of all these isolates was confirmed by the antimicrobial susceptibility test (Table 3).

In this study, ESBL genes were found only in the feces of gulls, *bla*_CTX-M-1*5*_ the most prevalent (3/7, 42.9%), followed by *bla_SHV-12_* and *bla_SHV-55_* (both present in 2/7, 28.6%) and *bla*_CTX-M-*1*_ (in 1/7, 14.3%). Resfinder also identified an *ampC*-promoter in the EAEC isolate (Ec_P17-I) associated with cephalosporins’ resistance, although no *amp*C gene was found. In this particular case, the assembly was then submitted to CARD “https://card.mcmaster.ca/home (accessed on 10 November 2022)”, and the gene *bla*_EC-14_ type (EC β-lactamase) was identified. Two of the seven ESBL producers (28.6%) contained more than one ESBL gene, *bla*_CTX-M-15_ and *bla*_SHV-55_.

All *Salmonella* and *Campylobacter* strains studied were phenotypically susceptible to the tested antimicrobials, except one strain of *C. jejuni* presenting, resistance to Ciprofloxacin, Tetracycline, and Ampicillin simultaneously, and harboring the genetic determinants *gyrA_T86I*, *tetO*, and *bla*_OXA-466_ in accordance with the MDR phenotype.

Genomic analysis of *S*.60:k:e,n,x,z15, *S*. Typhimurium, and *S*. Litchfield revealed the presence of the *aac(6’)-Iaa gene* in these isolates. 

## 4. Discussion

Data from several authors concerning the prevalence values for pathogenic *E. coli*, *Campylobacter* spp., and *Salmonella* spp. on fecal samples collected from wild birds in Europe are highly variable. Pathogenic *E. coli* was detected in about 6 to 30% of the samples [47,48,49], *Campylobacter* spp. incidence is reported to be between 1 and 50% [15,16,17,18,20,21,22,48,50,51,52] and *Salmonella* between 0 and 53% [16,17,18,19,21,22,48].

Prevalence estimates may diverge between studies due, among other factors, to different sampling procedures and detection methods, which may differ in sensitivity. Moreover, it is known that the probability of *Campylobacter* spp. and *E. coli* isolation from the microbiome of different wild bird species may have seasonal fluctuations and be influenced by the feeding habits of the birds, body mass, migratory behavior, as well as by their age, and nutritional conditions [20,48,51,52]. The factors influencing *E. coli*, *Campylobacter* spp., and *Salmonella* spp. prevalence values in wild birds are complex and may be linked to ecological and phylogenetic factors. Unless such factors are carefully considered, comparisons of prevalence values between studies may be ambiguous and not reliable. Nevertheless, it should be noted that in agreement with the work of Waldenström et al. [51], in our study, eight of the nine *Campylobacter* spp. strains were isolated from fecal samples of birds belonging to ecological guilds linked to higher *Campylobacter* spp. prevalence values. Four among the five *C. jejuni* isolates were found in fecal samples from four gulls (*Larus* spp.) which belong to the shoreline foraging invertebrate feeders guild, and three among the four *C. coli* and one *C. jejuni* strains were isolated from fecal samples of two eurasian coots (*Fulica atra*) (two *C. coli* isolates) and two common moorhens (*Gallinula chloropus*) (one *C. coli* and one *C. jejuni* MDR isolate), both aquatic invertebrate feeders. This may be of relevance since it is recognized that aquatic environments are one of the key reservoirs and transmission routes for the spread of pathogenic bacteria and of antimicrobial resistance [53]. 

As AMR poses a serious global threat of rising concern to the human, animal, and environmental health, due to the emergence, spread, and persistence of MDR bacteria, which undermine the effective use of the antimicrobial agents [54], AMR profile of all pathogenic isolates, as well as of a part of the commensal *E. coli*, was carried out.

Although antibiotic resistance is an ancient, naturally occurring phenomenon, widespread in the environment [55], it is evident that human activity has an impact on the evolution and mobilization of environmental antibiotic resistome [56]. The use and misuse of antimicrobials in human, animal, and environmental sectors led to the global emergence of reservoirs of resistance within and between these sectors and around the globe due to pollution by sewage, pharmaceutical industry waste, and manure runoff from farms [57] as well as the international trade of animals and food, and long-distance migratory patterns of wildlife.

Extended-spectrum β-lactamases (ESBLs), a group of enzymes conferring resistance to third-generation cephalosporins, have rapidly increased in *Enterobacteriacae*, constituting a major challenge to human health care.

Due to their close interaction with human-influenced environments and, to a large extent, feed on human food-animal waste, as well as their vast global distribution, gulls are birds particularly exposed to human waste and sewage. Thus, unsurprisingly, the carriage of ESBL-producing *E. coli* by gulls has already been documented by several other authors [24,29,30,58,59,60]. When comparing the ESBL types found in our study with the ones described in other Portuguese studies focused on gulls, we observe that, in accordance with our results, CTX-M-15 is the most prevalent ESBL found in gulls from the Oporto region [29] and was also identified in gulls from the Lisbon district [30]. Furthermore, *_blaCTX-M-1_* was also identified in the Oporto study and in gulls from Berlengas [24]. Other previously found ESBL enzymes (OXA-48, OXA-181, CTX-M-14-a, CTX-M-32, CTX-M-9, and TEM-52) [24,29,30] were not identified in our gull population. Noteworthy, *bla_SHV-12_* and *bla_SHV-55_* here detected in two gulls were not reported in the other Portuguese studies. As observed in previous studies, the ESBL strains detected in this study showed similarities with human clinical isolates. CTX-M-15, identified in three ExPEC isolates, is the predominant ESBL found among *Enterobacterioaceae* strains with clinical significance in Portugal [61,62,63] and worldwide [64]. SHV-12 and CTX-M-1, also found in two and in one ExPEC isolate, respectively, were also already frequently identified among Portuguese human isolates [65].

The detection of AMR MDR and of ESBL producing *E. coli* detected in our work is in accordance with other previous studies from wild birds in Portugal [23,24,25,26,27,28,29,30] and highlights the need for efforts to improve controls on the use of medically important antimicrobials, as well on the environmental transmission via pollution from several sources (e.g., industrial, residential, health care facilities, animal and plant production). These challenges and the consequent implementation of effective and adequate strategies will be critical to minimize the risk of exposing wildlife to human waste and sewage to prevent and tackle further contamination and dissemination of antibiotic resistance. 

Based on *E. coli* typing results, it is important to mention that most of the identified *E. coli* serotypes were already found to be associated with human diseases, such as the O92:H33 serotype isolated from stool samples of patients examined in the frame of clinical indications [66]; O111:H21 associated to a household outbreak in Northern Ireland [67]; two *E. coli* O45:H2 associated to outbreaks in the United States that have caused 18 cases of illness [68]; O125ac:H6 isolated from cases of diarrhea in Brazil, Germany and Australia [69]; O55:H10 already associated to cases of Hemolytic Uremic Syndrome (HUS) [70] and *E. coli* O25:H4- ST131 isolated from urine culture of a patient undergoing chronic hemodialysis at a tertiary care center in Chennai, India, in 2014 [71]. O105:H7 and ESBL O101:H9 were also already detected in human stool samples [72,73].

Most of the identified *E. coli* STs (ST10, ST20, ST23, ST34, ST40, ST453, and ST583) were already documented in the MLST database of diarrheagenic *E. coli* isolates collected from 1950 to 2015 in 44 countries distributed across Asia, America, Africa, and Europe [74]. Furthermore, ST10, ST23, and ST131 appear on the top 20 ExPEC STs recovered from human extraintestinal infections or of the gut [75]. Indeed, ST131 is the most frequent and disseminated Sequence Type among MDR ExPEC isolates worldwide [76].

Concerning *Salmonella* isolates characterization and its potential relation with human infection, it is important to notice that *Salmonella enterica* subsp. *enterica* serovars Typhimurium and Litchfield are important causative agents of human gastroenteritis and bacteremia in many countries. Over 2700 *Salmonella* serovars have been identified, being *S*. Typhimurium the most commonly associated with human and animal disease globally. The ST19, herewith detected, has been established as the founder and the most prevalent worldwide ST within *S. enterica* serovar Typhimurium [77]. *S*.60:k:e,n,x,z15 belong to *Salmonella enterica* subsp. *diarizonae*. This *Salmonella* subspecies is usually found in cold-blooded animals and in the environment. In fact, *S*.60:k:e,n,x,z15 has been isolated from stool samples of Brazilian snakes (*Crotalus durissus*) [78]. Curiously, in the present study, this *Salmonella* serovar was found in stool samples of a short-toed snake eagle (*Circaetus gallicus*), a bird that feeds almost exclusively on snakes. This fact reinforces the idea that birds may potentially be involved in the transport and dissemination of pathogenic microorganisms. 

The two *Campylobacter* species detected in this study (*C. jejuni* and *C. coli*) are the most frequently reported in human infections [79]. However, the corresponding STs are rarely described according to the *Campylobacter jejuni*/*coli* PubMLST database (as of October 2022). Nevertheless, regarding *C. jejuni*, ST990 is documented to have been found in strains isolated from humans with gastroenteritis and in chicken, also exhibiting an MDR (CIP, TET, AMP) profile similar to these sources; ST1268 was also already isolated from wild birds, environmental waters, goose, food, and humans with gastroenteritis, while ST8572 was described once in a wild bird in South Africa. The two ST from *C. coli*, ST11400 and ST11401, have never been described before. It is important to notice that although *C. jejuni*, *C. coli*, and *C. lari* are the three common *Campylobacter* species isolated from wild birds in Europe [20,50], *C. lari* was not detected in this study. Regarding virulence traits, besides the ones which are ubiquity in *C. jejuni* and *C. coli*, such as the ones encoding for toxins, flagella, and capsular polysaccharides, the secretion systems T4SS and T6SS seem to contribute to enhanced virulence and bacterial survival in isolates expressing these systems. T4SS, coded in plasmid pVir, is reported to be associated with adherence and invasion [80,81]. Regarding *Campylobacter* T6SS, previous studies demonstrated its involvement in erythrocyte cytotoxicity, host cell adherence, and colonization [82,83]. In the present study, among the isolates analyzed by WGS, both *C. coli* and one out of three *C. jejuni* harbor a complete T6SS, with a *C. coli* isolate being also positive for T4SS in *pVir* plasmid, suggesting that these isolates likely have an enhanced pathogenic potential and pose an increased risk to human health.

## 5. Conclusions

In conclusion, our results showed that free-living birds in Portugal act as carriers of *E. coli*, *Campylobacter* spp., and *Salmonella* spp. genotypes already associated with human disease, some of them resistant and multidrug-resistant to antimicrobial agents, including critically important antimicrobials for human medicine. These birds may act as reservoirs and effective dispersers of pathogens via fecal contamination (of food plants, irrigation water, soils, etc.) due to their ability to cover long distances during annual movements, with significant implications for public health management. These results reinforce the already recognized One Health concept showing that human health and animal health are interdependent and bound to the health of ecosystems in which they exist.

## Figures and Tables

**Table 1 ijerph-20-00223-t001:** Isolation and characterization of *E. coli*, *Salmonella* spp., and *Campylobacter* spp. in 115 fecal samples.

	Birds Classification		#*	*Escherichia coli* Isolates					*Salmonella* Isolates	*Campylobacter* Isolates
	Order	Species (Total Samples Analysed)		PCR-Based			WGS-Based	Non-Pathogenic		
				STEC	EAEC	EPEC	ExPEC			
EVOA	*Anseriformes*	*Anas crecca* (17)	17/17/10	1	0	0	0	15	0	0
		*Anas platyrhynchos* (1)	1/1/1	0	0	0	0	1	0	0
	*Ciconiiformes*	*Ixobrychus minutus* (1)	1/1/1	0	0	0	0	0	0	0
	*Gruiformes*	*Fulica atra* (2) **	2/2/2	0	0	0	0	2	0	2 *C. coli*
		*Gallinula chloropus* (7) **	7/7/5	0	1	0	1	5	0	1 *C. coli/*1 *C. jejuni*
	*Passeriformes*	*Acrocephalus schoenobaenus* (1)	1/1/1	0	0	0	0	0	0	0
		*Acrocephalus scirpaceus* (6) **	6/6/6	0	0	0	2	1	0	1 *C. coli*
		*Cettia cetti* (1)	1/1/1	0	0	0	0	0	0	0
		*Carduelis chloris* (1)	1/1/1	0	0	0	0	0	0	0
		*Emberiza calandra* (3)	3/3/3	0	0	0	0	2	0	0
		*Erithacus rubecula* (1)	1/1/1	0	0	0	0	0	0	0
		*Euplectes afer* (7)	7/7/7	0	0	0	0	0	0	0
		*Luscinia svecica* (6)	6/6/6	0	0	0	0	1	0	0
		*Oenanthe oenanthe* (1)	1/1/1	0	0	0	0	0	0	0
		*Passer domesticus* (1)	1/1/1	0	0	0	0	0	0	0
		*Phylloscopus collybita* (4)	4/4/4	0	0	0	0	1	0	0
		*Phylloscopus trochilus* (7)	7/7/7	0	0	0	0	1	0	0
		*Saxicola rubicola* (1)	1/1/1	0	0	0	0	0	0	0
		*Strunus unicolor* (3)	3/3/3	0	0	0	0	1	0	0
		Total (%)	71/71/62	1 (1.4)	1 (1.4)	0 (0)	3 (4.2)	30 (42.2)	0 (0)	5 (8.1)
CRFPBG	*Accipitriformes*	*Circaetus gallicus* (1) **	1/1/1	0	0	0	0	1	1 ^a^	0
	*Ciconiiformes*	*Ciconia nigra* (1)	1/1/1	0	0	0	0	0	0	0
	*Charadriiformes*	*Larus* spp. (18) **	18/18/14	0	0	1	0	10	0	4 *C. jejuni*
		*Larus fuscus* (1)	1/1/1	0	0	0	0	1	0	0
		*Fratercula arctica* (1)	1/1/1	0	0	0	0	0	0	0
	*Columbiformes*	*Columba livia* (7) **	7/7/6	1	0	0	0	5	2 ^b^	0
		*Streptopelia decaocto* (3)	3/3/2	0	0	0	0	3	0	0
	*Falconiformes*	*Falco peregrinus* (1)	1/1/1	0	0	0	0	0	0	0
	*Passeriformes*	*Garrulus glandarius* (1)	1/1/1	0	0	0	0	1	0	0
	*Pelecaniformes*	*Nycticorax nycticorax* (1)	1/1/0	0	0	0	0	1	0	0
	*Strigiformes*	*Athene noctua* (2)	2/1/1	0	0	0	0	2	0	0
		Total (%)	37/37/29	1 (2.7)	0 (0)	1 (2.7)	0 (0)	24 (64.9)	3 (8.1)	4 (13.8)
ICBAS	*Charadriiformes*	*Larus* spp. (50) ***	50/0/0	0	1	0	6	0	ND	ND
		Total (%)	50/0/0	0	1(2.0)	0 (0)	6 (12.0)	0 (0)	ND	ND
		Total (%)	158/108/91	2 (1.3)	2 (1.3)	1 (0.6)	9 (5.7)	54 (34.1)	3 (2.8)	9 (9.9)

#*, number of samples tested for *E. coli*/*Salmonella*/*Campylobacter*; **, more than one isolate per sample; ***, One pooled sample made up of 50 individual fecal droppings; ^a^, *S.*60:k:e,n,x,z_15_; ^b^, *S.* Typhimurium and *S.* Litchfield; ND, not determined; EVOA, Espaço de Visitação e Observação de Aves; CRFPBG, Centro de Recuperação de Fauna do Parque Biológico de Gaia; ICBAS, Instituto de Ciências Biomédicas Abel Salazar.

**Table 2 ijerph-20-00223-t002:** Distribution of serotypes, sequence types, pathotypes, and virulence determinants among the 14 pathogenic *E. coli* isolates, determined by PCR and WGS.

		EAEC		STEC		EPEC	ExPEC						
	*O antigen*	O92	O111	O105	O45	O125 ac	O101	O55	O78	O25	OND	OND	Total (%)
	*H antigen*	H33	H21	H7	H2	H6	H9	H10	H4	H4	H16	H9	
	*Sequence Type*	ST34	ST40	ST13581 ***	ST20	ST583	ST10	ST162	ST23	ST131	ST453	ST410	
	*# isolates*	1	1 **	1	1 *	1	3 **	1 **	1 **	2 **	1 **	1 **	14
Birds	*Acrocephalus scirpaceus*	0	0	0	0	0	2	0	0	0	0	0	2
	*Gallinula chloropus*	1	0	0	0	0	1	0	0	0	0	0	2
	*Anas crecca*	0	0	1	0	0	0	0	0	0	0	0	1
	*Columba livia*	0	0	0	1	0	0	0	0	0	0	0	1
	*Larus* spp.	0	1	0	0	1	0	1	1	2	1	1	8
Toxins	*astA*	0	1	0	1	0	3	0	0	0	0	0	5 (35.7)
	*cea*	0	0	0	0	0	3	0	0	0	0	0	3 (21.4)
	*cvaC*	0	0	0	0	0	0	1	1	0	1	0	3 (21.4)
	*mchF*	1	0	0	0	0	0	1	1	0	1	0	4 (28.6)
	*stx2*	0	0	1	1	0	0	0	0	0	0	0	2 (14.2)
Adhesins	*eae*	0	0	1	1	1	0	0	0	0	0	0	3 (21.4)
	*Iha*	1	1	0	0	0	0	0	1	2	0	0	5 (35.7)
	*lpfA*	0	1	1	0	0	0	1	1	0	1	1	6 (42.9)
	*papC*	0	0	0	0	0	0	1	0	1	1	0	3 (21.4)
	*yfcV*	0	0	0	0	1	0	0	0	2	0	0	3 (21.4)
siderophores	*fyua*	1	1	0	0	0	0	1	1	2	1	1	8 (57.1)
	*iroN*	0	1	0	0	0	0	1	1	0	1	0	4 (28.6)
	*irp2*	1	1	0	0	0	0	1	1	2	1	1	8 (57.1)
	*iucC*	1	1	0	0	0	0	1	1	2	1	1	8 (57.1)
	*iutA*	1	1	0	0	0	0	1	1	2	1	1	8 (57.1)
	*sitA*	0	0	0	0	0	0	1	1	2	1	1	6 (42.9)
Other virulence genes	*aggR*	1	1	0	0	0	0	0	0	0	0	0	2 (14.2)
	*chuA*	0	0	0	0	1	0	0	0	2	0	0	3 (21.4)
	*espA*	0	0	1	1	1	0	0	0	0	0	0	3 (21.4)
	*etsC*	0	0	0	0	0	0	1	1	0	1	0	3 (21.4)
	*gad*	1	1	1	1	1	3	1	1	2	1	1	14 (100)
	*hra*	0	0	0	1	0	0	0	1	1	0	0	3 (21.4)
	*hlyF*	0	0	0	0	0	0	1	1	0	1	0	3 (21.4)
	*iss*	0	1	1	1	0	3	1	1	2	1	0	11 (78.6)
	*kpsE*	0	0	0	0	0	0	0	0	2	1	0	3 (21.4)
	*nleB*	0	0	1	1	1	0	0	0	0	0	0	3 (21.4)
	*OmpT*	0	1	1	1	1	0	1	1	2	1	0	9 (64.3)
	*sat*	1	1	0	0	0	0	0	0	2	0	0	4 (28.6)
	*terC*	1	1	1	1	1	3	1	1	2	1	1	14 (100)
	*TraT*	0	1	1	1	0	3	1	0	2	1	0	10 (71.4)

*, Antimicrobial resistant (AMR) isolate; **, Multidrug resistant (MDR) isolate; ***, Novel sequence type; EAEC, Enteroaggregative *E. coli*; STEC, Shiga toxin-producing *E. coli*; EPEC, Enteropathogenic *E. coli*; ExPEC, Extraintestinal pathogenic *E. coli*; ST, Sequence type; OND, O antigen not determined.

**Table 3 ijerph-20-00223-t003:** Resistance patterns of pathogenic and non-pathogenic *Escherichia coli* isolates (*n* = 14) in the different bird species.

Bird Species	Pathotype	Serotype/ST	Resistance Phenotype	AMR Determinants (WGS)
*Anas crecca*	non-pathogenic	ND	AMP, TET	ND
*Larus fuscus*			AMC, AMP	
*Larus* spp.			AMP, TET	
*Larus* spp.	ExPEC	O55:H10/ ST162	AMP, CAZ, CIP, COX, FEP, NAL, SMX, TET, TMP	*bla*_TEM-1B,_*bla*_SHV-12,_*gyrA*, *qnrB19*_,_*sul2*, *tet(B)*, *dfrA17*
		ONH:H16/ ST453	AMP, CAZ, CIP, COX, FEP, NAL, SMX, TET, TMP	*bla*_TEM-1B,_*bla*_SHV-12,_*gyrA*, *qnrB19*_,_*sul2*, *tet(B)*, *dfrA14*
		OND:H9/ ST410	AMC, AMP, AZM, CAZ, CHL, CIP, COX, FEP, NAL, SMX, TET, TMP	*aac(6’)-Ib-cr*, *mph(A)*, *bla*_CTX-M-15,_*bla*_TEM-1A,_*bla*_OXA-1_, *floR*, *gyrA*, *sul1*, *sul2*, *tet(A)*, *dfrA17*
		O78:H4/ST23	AMP, CAZ, CIP, COX, FEP, NAL, SMX, TET	*bla*_CTX-M-1_, *gyrA*, *sul2*, *tet(A)*
		O25:H4/ ST131	AMC, AMP, AZM, CAZ, CIP, COX, FEP, GMN, NAL, SMX, TET, TMP	*aac(6’)-Ib-cr*, *mph(A)*, *bla*_CTX-M-15_, *bla*_SHV-55_, *bla*_OXA-1_, *aac(3)-IIa*, *gyrA*, *qnrS1, sul1*, *tet(A)*, *dfrA14*, *dfrA17*
		O25:H4/ ST131	AMC, AMP, AZM, CAZ, CIP, COX, FEP, GMN, NAL, SMX, TET, TMP	*aac(6’)-Ib-cr*, *mph(A)*, *bla*_CTX-M-15_, *bla*_SHV-55_, *bla*_OXA-1_, *aac(3)-IIa*, *gyrA*, *qnrS1, sul1*, *tet(A)*, *dfrA17*
*Larus* spp.	EAEC	O111:H21/ ST40	AMC, AMP, CAZ, COX, FOX	*ampC-promoter*, *bla*_TEM-1B_
*Acrocephalus scirpaceus*	ExPEC	O101:H9/ ST10	AMP, CHL, CIP, TET	*bla*_TEM-1B_, *cmlA1*, *qnrB19*, *qnrB82*, *qnrB67*, *qnrB56*, *tet(A)*
		O101:H9/ ST10	AMP, CHL, CIP, TET	*bla*_TEM-1B_, *cmlA1*, *qnrB19*, *qnrB82*, *qnrB67*, *qnrB56*, *tet(A)*
*Gallinula chloropus*	ExPEC	O101:H9/ ST10	AMP, CHL, CIP, TET	*bla*_TEM-1B_, *cmlA1*, *qnrB19*, *qnrB82*, *qnrB67*, *qnrB56*, *tet(A)*
*Columba livia*	STEC	O45:H2/ST20	SMX, TET, TMP	*sul1*, *tet(A)*, *dfrA1*

ND, not determined; ST, Sequence Type; EAEC, Enteroaggregative *E. coli*; ExPEC, Extraintestinal *E. coli*; STEC, Shiga toxin-producing *E. coli*; OND, O antigen not determined; AMP, Ampicillin; AMC, Amoxicillin-Clavulanic Acid; AZM, Azithromycin; FEP, Cefepime (FEP); COX, Cefotaxime; FOX, Cefoxitin; CZD, Ceftazidime; CRO, Ceftriaxone; CHL, Chloramphenicol; ERY, Erythromycin; GMN, Gentamicin; MEM, Meropenem; NAL, Nalidixic acid; CIP, Ciprofloxacin; SMX, Sulfamethoxazole; TET, Tetracycline; TGC, Tigecycline; TMP, Trimethoprim.

## Data Availability

All supporting data and protocols have been provided within the article or through supplementary data files. Supplementary Material is available with the online version of this article.

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
