# Peer review of "Genotypic and Phenotypic Characterization of Pathogenic Escherichia coli, Salmonella spp., and Campylobacter spp., in Free-Living Birds in Mainland Portugal"

_ijerph, 2022, doi:10.3390/ijerph20010223_

Round 1
Reviewer 1 Report
Introduction. L75. Many other studies have been performed in wild birds in Portugal. Include more information on these studies, namely which birds, regions, comparing with the birds and regions focussed in the present study.
L89: It should be clear that the 2 study areas include 3 recovery areas. It is not clear for the international reader that Vila Nova de Gaia and Oporto are located in the north of Portugal
L97: it should be Centro de Recuperação de Fauna do Parque Biológico de Gaia to be in agreement with Table 1.
L108: the abbreviation should be CRFPBG as in Table 1
Additional information should be included in M&M:
- What species are included in "wild birds"
- Clarify "recovered animals". Also fresh dropping feces?
- The period of recovery of samples from ICBAS was never mentioned. What was the rationale to include these samples since the methodology was not the same as for the remaining collection? Is the period of isolation compatible with the remaining collection?
- L118: what was the selective medium for E. coli?
- L122: clarify selective enrichment
- L144: why using meropenem instead of ertapenem since some carbapenemase producers (OXA-48 type) often show susceptibility to meropenem but are resistant to ertapenem.
- L191: why is not the denominator always 158? It is 108 for Salmonella and 91 for Campylobacter. Clarify
- Table 3: The 3 non pathogenic E. coli need to be typed by WGS. Some genes are not in italic (blaOXA-1 for OND:H9 / ST410 ). What is "ampC-promotor"? Does this isolate harbor an ampC gene? which one?
- L272: replace ESBL mechanisms by ESBL types
- L276: indicate which blaTEM, blaSHV, etc since not all are ESBLs
Discussion: comment on the differences and resemblances found with previous studies from wild birds in Portugal
Author Response
Dear Reviewer,
We deeply appreciate your corrections, comments, and suggestions, which will improve the quality of our manuscript. Thank you for that.
Below you will find our explanations and considerations on your revision:
Introduction. L75. Many other studies have been performed in wild birds in Portugal. Include more information on these studies, namely which birds, regions, comparing with the birds and regions focussed in the present study.
Answer: We have introduced more references regarding studies in wild birds in Portugal in introduction (line 79 of revised manuscript) and add one sentence in discussion mentioning some of the similarities and differences presented in some of these studies with the one here presented (lines 354-361).
L89: It should be clear that the 2 study areas include 3 recovery areas. It is not clear for the international reader that Vila Nova de Gaia and Oporto are located in the north of Portugal
Answer: We have tried to clarify this information, please see lines 94-95 and 120 of the revised manuscript.
L97: it should be Centro de Recuperação de Fauna do Parque Biológico de Gaia to be in agreement with Table 1.
Answer: We have clarified this information (lines 103-104)
L108: the abbreviation should be CRFPBG as in Table 1
Answer: Done (line 114).
Additional information should be included in M&M:
- What species are included in "wild birds"
Answer: We have included, in M&M, the number of tested species and referred table 1 (lines 110-111). We have substitute the sentence “wild birds” by “free-living birds”.
- Clarify "recovered animals". Also fresh dropping feces?
Answer: We have changed the word “recovered” to “hosted” (line 114). The animals in the study were animals that were in CRFPBG due to the reasons presented in M&M (injured, debilitated or illegally kept). Furthermore, we have decided to substitute “fresh dropping feces” by “fresh feces” (line 111).
- The period of recovery of samples from ICBAS was never mentioned. What was the rationale to include these samples since the methodology was not the same as for the remaining collection? Is the period of isolation compatible with the remaining collection?
Answer: Yes, the period of isolation of ICBAS samples is compatible with the remaining collection and is mentioned on line 119 of the revised manuscript. Yes, it is true that ICBAS has a different methodology regarding sampling, however, we think that ICBAS samples are of great interest for this study and should not be neglected. That’s why we included them. The use of one pooled sample made up of 50 fecal dropping from gulls of Matosinhos beach is related with the fact that this sample was collected from birds that live in large flocks and we wanted to avoid capturing specimens. Evidently, there is an asymmetry, but it results from ecological characteristics (inherent to birds) and has an incisive purpose, that is to represent in this study the largest possible number of species.
- L118: what was the selective medium for E. coli?
Answer: For E. coli we have used Tryptone Bile X-Glucuronide (TBX, Biokar Diagnostics, Pantin, France) and CHROMagar STEC (CHROMagar, Paris, France), both mentioned on now reference 35. We had to remove this information from the manuscript as recommended by the editors in order to avoid repetition from our previous paper (reference 35).
- L122: clarify selective enrichment
Answer: For Salmonella, after a pre-enrichment step, in Buffered Peptone Water (1:10 dilution; BPW-Oxoid, Basingstoke, Hampshire, UK) at 37 °C overnight, a selective enrichment of 1 mL culture was performed in 10 mL of Muller–Kauffmann tetrathionate–novobiocin broth (MKTTn, Biokar Diagnostics) at 37 °C for 24 h. This methodology is fully described in reference 35 and is absent in this manuscript for the reason presented above.
- L144: why using meropenem instead of ertapenem since some carbapenemase producers (OXA-48 type) often show susceptibility to meropenem but are resistant to ertapenem.
Answer: According to EUCAST, for carbapenemase screening a meropenem screening cut-off of >0.125 mg/L (zone diameter <28 mm) is recommended.
- L191: why is not the denominator always 158? It is 108 for Salmonella and 91 for Campylobacter. Clarify
Answer: The denominator is equal to the number of samples tested for each of the microorganisms. We have tested 158 samples for E. coli, 108 for Salmonella spp. and 91 for Campylobacter spp. Please see Table 1.
- Table 3: The 3 non pathogenic E. coli need to be typed by WGS. Some genes are not in italic (blaOXA-1 for OND:H9 / ST410 ). What is "ampC-promotor"? Does this isolate harbor an ampC gene? which one?
Answer: In this work, we have decided to sequence only pathogenic and MDR isolates, that is why non-pathogenic E. coli isolates on Table 3 were not sequenced. We have changed blaOXA-1 to italic. We also clarified the ampC-promotor question in the Antimicrobial Resistance results (section 3.3), as follows: “Resfinder also identified an ampC-promoter in the EAEC isolate (Ec_P17-I) associated with cephalosporins’ resistance, although no ampC gene was found. In this particular case, the assembly was then submitted to CARD (https://card.mcmaster.ca/home) and the gene blaEC-14 type (EC β-lactamase) was identified.“ (strict match) (lines 285-289)
- L272: replace ESBL mechanisms by ESBL types
Answer: The sentence was replaced.
- L276: indicate which blaTEM, blaSHV, etc since not all are ESBLs
Answer: The reviewer is correct, and we really appreciated this comment. We have changed our mistake across all the manuscript. Indeed, blaTEM-1A, TEM-1B and blaOXA-1 are not considered as ESBL. For that reason, we changed the number of ESBL E. coli isolates (7 instead of 10 isolates, all of them in gulls), as well as all the related-percentage.
Discussion: comment on the differences and resemblances found with previous studies from wild birds in Portugal
Answer: We have mentioned the similarities of our study with the others from wild birds in Portugal in discussion (lines 354-361)
Reviewer 2 Report
The study concerns a current topic. The data provided by the authors in this article can increase knowledge about the possibility of diffusion of pathogenic bacteria, ARB and ARG in the environment by spreading with the feces of wild birds.
Specific comments
Abstract
Lines 26 what do the authors mean by “agricultural health”?
Line 29 author should specify that they analyse E.coli and pathogen E.coli
Lines 34-35 Authors should specify to what “ten of these refers”
Introduction
Lines 67 the transmission = the human transmission
Line 77 from living birds from different regions of Portugal.
Line 81 links between humans, livestock, and natural environments=links among humans, livestock, and natural environments
Materials and Methods
2.2. Studied population and sample collection
Author should improve the information on sampling:
· How the faeces of different birds were recognized?
· How long had the stool defined as “fresh” been laid?
· How was the “pooled” sample collected? Was it fresh?
2.3. Isolation Methodology
Lines 117-118 the authors should specify that they have carried out the research of E.coli and pathogenic E.coli and the specific modalities.
2.4. Typing and Antimicrobial Susceptibility Testing (AST)
Lines 155-158 authors should describe the method used for the identification of Campylobacter species
Results
3.2. Serotyping, virulence genes and MLST analysis
Authors should describe in more detail the results obtained in Table 2
3.3. Antimicrobial Resistance
Line 273 (6/10-60%)= (60%)
Discussion
Author should explain the implications of the results discussed in lines 308-315.
Conclusion
Authors should add some consideration on the role of wild bird faeces spread of human pathogens and the agriculture involvement in the introduction and in the conclusion paragraphs
Author Response
Dear Reviewer,
We deeply appreciate your corrections, comments, and suggestions, which will improve the quality of our manuscript. Thank you for that.
Below you will find our explanations and considerations on your revision:
Abstract
Lines 26 what do the authors mean by “agricultural health”?
Answer: Here, we intended to reinforce the concept of One Health, human health is interconnected with the health of the agricultural systems from where we obtain our food. However, in order to fulfill the reviewer’s concern we have changed this sentence to “… affecting human and agriculture” (line 28)
Line 29 author should specify that they analyse E.coli and pathogen E.coli
Answer: We agree with the reviewer and introduced this information on abstract (line 31).
Lines 34-35 Authors should specify to what “ten of these refers”
Answer: We are referring to E. coli isolates. We have tried to clarify this information (line 37).
Introduction
Lines 67 the transmission = the human transmission
Answer: Altered accordingly (line 70).
Line 77 from living birds from different regions of Portugal.
Answer: Altered accordingly (line 81).
Line 81 links between humans, livestock, and natural environments=links among humans, livestock, and natural environments
Answer: Altered accordingly (line 85).
Materials and Methods
2.2. Studied population and sample collection
Author should improve the information on sampling:
- How the faeces of different birds were recognized?
- How long had the stool defined as “fresh” been laid?
- How was the “pooled” sample collected? Was it fresh?
Answer: The feces of EVOA and CRFPBG were collected fresh and right after defecation. Matosinhos beach sample was taken at dawn, when the gulls are still on the beach and processed within 2h of sampling. We have tried to complete information regarding sampling on M&M (lines 112, 120-123).
2.3. Isolation Methodology
Lines 117-118 the authors should specify that they have carried out the research of E.coli and pathogenic E.coli and the specific modalities.
Answer: The description of how we identified pathogenic E. coli is already specified on item “2.4 Typing and Antimicrobial Susceptibility Testing (AST)”: ”Pathogenic E. coli identification was based on the presence of the most common virulence genes (VG), eae, aggR, elt, estp, and ipaH, performed by multiplex PCR (modified from [33-35]) and on the presence of Shiga toxins genes stx1 and stx2”. Furthermore, some Pathogenic E. coli isolates (ExPEC pathotype) were identified after sequencing of E. coli multidrug resistant (MDR) isolates. This information is also present on the manuscript (lines 207-210 of the revised manuscript). On Table 1 readers can clearly see which pathogenic E. coli were classified based on PCR and which were based on WGS (please see table 1 headline).
2.4. Typing and Antimicrobial Susceptibility Testing (AST)
Lines 155-158 authors should describe the method used for the identification of Campylobacter species
Answer: We have clarified in the methods section that all Campylobacter suspected colonies were confirmed at genus and species level by the oxidase activity test and MALDI-TOF mass spectrometry, respectively (VITEK® MS, bioMérieux) (line137).
Results
3.2. Serotyping, virulence genes and MLST analysis
Authors should describe in more detail the results obtained in Table 2
Answer: We have described in more detail some of the information presented on Table 2 (lines 232-234)
3.3. Antimicrobial Resistance
Line 273 (6/10-60%)= (60%)
Answer: This sentence was altered.
Discussion
Author should explain the implications of the results discussed in lines 308-315.
Answer: We have tried to follow this reviewer suggestion (lines 332-334 of the revised manuscript).
Conclusion
Authors should add some consideration on the role of wild bird faeces spread of human pathogens and the agriculture involvement in the introduction and in the conclusion paragraphs
Answer: We have tried to mention the agriculture involvement in the interface between wild birds and human in introduction and conclusions (lines 71-72 and 432-433).
Reviewer 3 Report
The article “The Birds are potential carriers of pathogens affecting human and agricultural health” investigate the prevalence of pathogenic strain of foodborne pathogens i.e., Escherichia coli, Salmonella spp. and Campylobacter spp in free-living. The study is interesting as it show the avian isolates have genetic association with human pathogenic strains highlighting their role in spread of virulent strains of bacteria. The article is well written although a couple of points need to be addressed to further improve the quality of paper.
Comments:
1. Line 112-113 what was the rationale behind using “One pooled sample made 112 up of 50 individual fecal droppings was collected in July 2021 from Matosinhos beach 113 (Oporto coastline) using a sterile glove.
2. Was there any statistical association of the bacteria with specific species of birds
3. How are the results shown in this study different/similar from/to already reported data in free living birds please elaborate in discussion section?
Author Response
Dear Reviewer,
We deeply appreciate your corrections, comments, and suggestions, which will improve the quality of our manuscript. Thank you for that.
Below you will find our explanations and considerations on your revision:
- Line 112-113 what was the rationale behind using “One pooled sample made up of 50 individual fecal droppings was collected in July 2021 from Matosinhos beach (Oporto coastline) using a sterile glove.
Answer: The use of one pooled sample made up of 50 fecal dropping from gulls of Matosinhos beach is related with the fact that this sample was collected from birds that live in large flocks and we wanted to avoid capturing specimens. Evidently, there is an asymmetry, but it results from ecological characteristics (inherent to birds) and has an incisive purpose, that is to represent in this study the largest possible number of species.
- Was there any statistical association of the bacteria with specific species of birds
Answer: In this study we have tested feces from at least 30 different bird species, in some cases we only got one sample for each specimen and sometimes not enough sample to test all three microorganisms (E. coli, Salmonella spp. and Campylobacter spp). Furthermore, in the case of Matosinhos sample only E. coli was tested. We do not think a statistical association of bacteria with specific species of birds would be possible or accurate in this study.
- How are the results shown in this study different/similar from/to already reported data in free living birds please elaborate in discussion section?
Answer: We have introduced a sentence mentioning the similarities of our study with the others from wild birds in Portugal in discussion (lines 354-361)